# Genome-Wide Analysis of Type-III Polyketide Synthases in Wheat and Possible Roles in Wheat Sheath-Blight Resistance

**DOI:** 10.3390/ijms23137187

**Published:** 2022-06-28

**Authors:** Xingxia Geng, Yihua Chen, Shufa Zhang, Zhen Gao, Shuhui Liu, Qunhui Yang, Jun Wu, Xinhong Chen

**Affiliations:** Shaanxi Key Laboratory of Genetic Engineering for Plant Breeding, College of Agronomy, Northwest A&F University, Xianyang 712100, China; gengxingxia@126.com (X.G.); cyh199733@126.com (Y.C.); zhangshufa1362@163.com (S.Z.); gaozhen2019@163.com (Z.G.); liushuhui830@163.com (S.L.); yangh8998@163.com (Q.Y.); 13572016162@163.com (J.W.)

**Keywords:** genome-wide analysis, *TaPKS*, wheat (*Triticum aestivum* L.), sheath blight, *Rhizoctonia cerealis*

## Abstract

The enzymes in the chalcone synthase family, also known as type-III polyketide synthases (PKSs), play important roles in the biosynthesis of various plant secondary metabolites and plant adaptation to environmental stresses. There have been few detailed reports regarding the gene and tissue expression profiles of the *PKS* (*TaPKS*) family members in wheat (*Triticum aestivum* L.). In this study, 81 candidate *TaPKS* genes were identified in the wheat genome, which were designated as *TaPKS1–81*. Phylogenetic analysis divided the *TaPKS* genes into two groups. *TaPKS* gene family expansion mainly occurred via tandem duplication and fragment duplication. In addition, we analyzed the physical and chemical properties, gene structures, and cis-acting elements of *TaPKS* gene family members. RNA-seq analysis showed that the expression of *TaPKS* genes was tissue-specific, and their expression levels differed before and after infection with *Rhizoctonia cerealis*. The expression levels of four *TaPKS* genes were also analyzed via qRT-PCR after treatment with methyl jasmonate, salicylic acid, abscisic acid, and ethylene. In the present study, we systematically identified and analyzed *TaPKS* gene family members in wheat, and our findings may facilitate the cloning of candidate genes associated with resistance to sheath blight in wheat.

## 1. Introduction

Plants resist environmental pressures by producing corresponding secondary metabolites after interacting with the environment. Polyketide-derived compounds such as flavonoids, anthrone, resorcinol, quinolones, and chromones are natural secondary metabolites in plants with diverse biological functions [1]. Polyketides are usually produced via the action of polyketide synthases (PKSs). According to the protein structure, PKSs can be divided into types I, II, and III [2,3,4,5,6]. Type-I and -II PKSs exist in microorganisms. Type-III PKSs in the chalcone synthase (CHS) superfamily are mainly distributed in plants, but also in microorganisms. Type-III PKSs in the CHS superfamily comprise 40–45 kDa dimer polypeptides, which can complete repeated condensation reactions. The coding gene usually comprises two exons and one intron. The highly conserved catalytic active center (the combined structure of Cys–His–Asn) is exactly the same in all known type-III *PKS* genes, and it is responsible for catalyzing the formation of the main chains of various polyketide compounds [1,7]. The functional differences in plant type-III PKSs are mainly characterized by the choice of initial substrate and the number of condensation reactions (polyketone chain extension length) [8,9,10]. In particular, CHS catalyzes the condensation of p-coumaroyl-CoA and three molecules of malonyl-CoA through three enzymatic reactions to produce enzyme-bound tetrakis ketone compound intermediates, before undergoing Claisen-type cyclization to form naringenin chalcone (Figure 1a); this is an important intermediate in the biosynthesis of plant flavonoids and isoflavones [1]. In addition, 2-pyrone synthase uses acetyl-coenzyme A as the starting substrate and condenses with two malonyl-CoA molecules to produce triacetic acid lactone [11,12] (Figure 1b). Plant type-III PKSs are responsible for synthesizing important secondary metabolites in plant defense and signal transduction processes. The expression of *PKS* genes can be induced by various biotic and abiotic stimuli, including ultraviolet radiation, hormones, or a pathogen attack [13,14]. For example, the expression of the *CHS* gene increased in *Plagiochasma appendiculatum* after treatment with hormones comprising abscisic acid, salicylic acid, and methyl jasmonate [1]. The induced expression of the stilbene synthase gene also varied in grapes with different degrees of resistance to *Plasmopara viticola* infection [15].

Wheat is the third most important cereal crop in the world after rice and corn, and about 35% of the world’s population is reliant on wheat [16]. Wheat sheath-blight caused by the necrotrophic fungus *Rhizoctonia cerealis* is highly destructive. This disease has become an important limiting factor that affects wheat yields in China, and the yield losses can even exceed 50% in severe cases [17]. *Rhizoctonia cerealis* is a soil-borne fungal disease that can damage the leaf sheath and stem tissues of plants during interaction with the host, thereby affecting the transport of nutrients and reducing the yield [18,19]. *PKS* family genes play key roles in plant resistance and signal transduction, but little is known about the *PKS* gene family in wheat.

In this study, we conducted a systematic and comprehensive analysis of the wheat *PKS* gene family members for the first time; this included identifying the *TaPKS* family members and their phylogenetic relationships with those in other representative species, as well as assessing the structure and evolution of *TaPKS* genes, gene duplications, gene localizations, and gene promoters. In addition, we combined RNA-seq data for *Rhizoctonia cerealis* infection-resistant and -susceptible wheat materials to analyze the trends in the expression of *TaPKS* genes, and verified the expression of differentially expressed *TaPKS* genes via qRT-PCR. Moreover, qRT-PCR was conducted to analyze the expression patterns of differentially expressed *TaPKS* genes following treatment with salicylic acid, methyl jasmonate, abscisic acid, and ethylene. Our results provide an important reference for further evolutionary and functional studies of *PKS* genes in wheat, as well as facilitating the selection of suitable candidate genes associated with resistance to wheat sheath-blight.

## 2. Results

### 2.1. Genome-Wide Identification of PKS Genes in Wheat

In total, 98 wheat *PKS* genes were identified by searching the Ensembl Plants database using Pfam ID (PF00195 and PF02797), but 17 were rejected due to a lack of complete conserved domains, or duplicates. The numbers of amino acids in the gene are in the range of 334–478 and their molecular weights are 36.05–52.04 kDa. The isoelectric points calculated for the *TaPKS* gene products range from 5.25 to 6.9. Most *TaPKS* family genes are hydrophobic. Details of the predicted physical and chemical properties of the gene products are listed in Appendix A.

### 2.2. Phylogenetic Analysis of PKS Genes in Wheat

In order to systematically analyze the evolutionary relationships among wheat *PKS* family genes in different species, we constructed a phylogenetic tree by combining 81 wheat PKSs and 12 previously reported plant PKS superfamily polypeptide sequences (Figure 2). Previous studies have shown that members of the PKS superfamily do not form a species-specific cluster, but instead, fall into two subfamilies comprising chalcone and non-chalcone subfamilies based on their function. The evolutionary tree in Figure 2 shows that the wheat *PKS* family genes were divided into CHS and non-CHS groups, with 25 *PKS* genes in the CHS group and 56 in the non-CHS group (Figure 2).

### 2.3. Chromosome Locations, Gene Duplications, and Synteny Analysis for TaPKS Genes

The chromosome mapping results showe that *TaPKS* family genes were distributed on all chromosomes, except for chromosomes 3A and 3B, and most were distributed at the ends of chromosomes in the form of gene clusters (Figure 3).

Tandem duplication and fragment duplication are essential for the evolution of gene families. Eight pairs of tandem duplication genes were identified in this study. In addition to three pairs of tandem gene clusters on Chr1B (*TaPKS04* and *TaPKS05*, *TaPKS05* and *TaPKS06*, and *TaPKS06* and *TaPKS07*), one pair is present on each of Chr2B (*TaPKS28* and *TaPKS29*), Chr2D (*TaPKS36* and *TaPKS37*), Chr4A (*TaPKS51* and *TaPKS52*), Chr5A (*TaPKS56* and *TaPKS57*), and Chr5D (*TaPKS63* and *TaPKS64*). Thirty-three pairs of segmental duplication genes were identified, whereby 30 segmental duplications occurred between partially homologous chromosomes and three between non-homologous chromosomes (Figure 4). 

In addition, we analyzed the Ka/Ks ratios for *PKS* gene pairs with tandem and fragment duplications to explore the evolutionary selection of *PKS* family genes in wheat (Appendix A). The Ka/Ks ratios determined for all duplicated genes were less than 1, thereby indicating that the *PKS* family genes in wheat have been subjected to strong purifying selection pressure to maintain their functional stability during the evolutionary process.

The collinearity coverage of the genome is greater when species are more closely related. To further infer the evolutionary origin and orthologous relationships among wheat *PKS* gene family members, we performed synteny analysis between wheat and the representative species comprising *Triticum dicoccoides*, maize, and rice (Figure 5). Figure 5 shows that the numbers of orthologous gene pairs in *T. dicoccoides*, maize, and rice were determined as 49, 4, and 3, respectively. The *TaPKS* genes are distantly related to the *PKS* genes in maize and rice, and only four and three *PKS* genes are in the syntenic segments of their genomes, respectively. By contrast, the *PKS* genes in wheat share a higher level of synteny with the *PKS* genes in *T. dicoccoides*, i.e., more genes are in synteny (49). Thus, the *PKS* genes in wheat and *T. dicoccoides* are closely related. 

### 2.4. Structure and Conserved Domains in TaPKS Genes

Gene structure analysis can identify gene functions and evolutionary relationships, so we used CDS and DNA genome sequences to predict the structures of the *TaPKS* genes. As shown in Figure 6, most *PKS* genes contain 1–3 exons; 47 genes contain two exons and one intron (i.e., 58% of the total *TaPKS* genes), 13 genes contain a single exon (i.e., 16% of the total number of *TaPKS* genes), and six genes (*TaPKS22*, *TaPKS22*, *TaPKS24*, *TaPKS28*, *TaPKS71*, and *TaPKS76*) contain three exons, whereby *TaPKS76* contains three introns and the other five *TaPKS* genes contain two introns. Only *TaPKS56* contains five exons. Most of the genes contain an intron. Analysis of the protein motifs in the *TaPKS* genes (Figure 6) detected 10 conserved motifs designated as motif 1 to motif 10. Motif 1, motif 3, motif 5, motif 7, and motif 10 are conserved motifs found in all *TaPKS* gene family members, whereas several other motifs are relatively conserved and do not occur in all *PKS* family members. The *TaPKS* genes with close evolutionary relationships usually have similar compositions and numbers of motifs, thereby indicating that the *TaPKS* gene family is evolutionarily conserved, and this structural stability may be closely related to the involvement of *PKS* genes in many important biological processes in plants. 

### 2.5. Promoter Analysis in 81 TaPKS Genes

We also analyzed the promoter regions upstream of the gene CDS regions. In total, 52 cis-acting elements were identified in the 2000 bp region upstream of the CDS for each *TaPKS* gene (Appendix A). According to their functions, these cis-acting elements can be divided into the following four categories (Figure 7): development-related elements with six types of elements; environmental-stress-related elements with four types of elements; hormone-responsive elements with nine types of elements; and light-responsive elements with thirty-three types of elements. Among the hormone-responsive-elements (TGA-element, AuxRR-core, GARE-motif, P-box, TATC-box, CGTCA-motif, TGACG-motif, TCA-element, and ABA-responsive element (ABRE)), excluding the *TaPKS*12, *TaPKS*13, *TaPKS*44, *TaPKS*62, *TaPKS*66, and *TaPKS*72 genes, the remaining 75 genes contain an MeJA-responsive element (CGTCA-motif and TGACG-motif), i.e., 92.6% of all genes. The promoter regions of six *TaPKS* genes (*TaPKS*16/17/24/55/66/68) do not contain an ABRE. The light-responsive elements comprise most of the types of elements, and 76 genes contain G-box elements, i.e., 93.8% of all *TaPKS* family genes. In addition, the number of G-box elements present in different *TaPKS* family genes ranges from 0 to 12. Fifty and forty-eight genes contain a CAT-box element and Box 4 element, respectively. These findings indicate that members of this family play important roles in the plant hormone response and light response processes. In addition, the numbers, types, and locations of cis-acting elements vary among the different gene promoter regions, which may be related to the specific functions of the genes. The promoter elements in each gene are shown in Appendix A.

### 2.6. Expression Profiles of TaPKS Genes

The data used to analyze the tissue-specific expression patterns of *TaPKS* genes were obtained from published RNA-Seq data. We analyzed the expression profiles for 81 *TaPKS* genes in root, leaf, seedling, spike, and anther tissues from wheat. Figure 8 shows that 69 *TaPKS* genes were differentially expressed in various tissues. The spike, anther, root, seeding, and leaf tissues contained 42, 16, 8, 9, and 4 *TaPKS* family genes, respectively, with relatively high expression levels. These results indicate that the expression of *TaPKS* genes is tissue-specific. Ten *TaPKS* genes (*TaPKS*48, *TaPKS*10, *TaPKS*03, *TaPKS*13, *TaPKS*49, *TaPKS*47, *TaPKS*74, *TaPKS*16, *TaPKS*72, and *TaPKS*66) were highly expressed in all tissues, whereas the other 59 *TaPKS* genes were highly expressed only in a single tissue, and mostly not in other tissues. Twelve other genes were not expressed in any of the five tissues evaluated.

### 2.7. TaPKS Gene Expression Pattern under Rhizoctonia Cerealis Stress

In order to understand the responses of *TaPKS* genes to infection by *Rhizoctonia cerealis*, we used RNA-seq data for the resistant wheat material H83 and moderately susceptible wheat material 7182, inoculated with *Rhizoctonia cerealis* in the adult stage, to analyze the expression levels of *TaPKS* family genes at different inoculation time points (Figure 9). According to the results, the *TaPKS* gene expression profiles differed greatly between the resistant and susceptible materials (Figure 9A). When the two materials were not inoculated with *Rhizoctonia cerealis*, the expression patterns of the *TaPKS* genes were generally consistent in the two materials. At 36 h after inoculation, the expression levels of the three *TaPKS* family genes differed significantly in the susceptible material 7182 compared with the uninfected control. In particular, the expression level of *TaPKS31* was upregulated, whereas the expression levels of *TaPKS68*, *TaPKS33*, and *TaPKS71* were downregulated. At this time, eight *TaPKS* family genes were differentially expressed in the disease-resistant material H83 compared with the susceptible material; *TaPKS*70, *TaPKS72*, and *TaPKS66* were significantly downregulated, whereas *TaPKS68*, *TaPKS71*, *TaPKS31*, *TaPKS29*, and *TaPKS33* were significantly upregulated. At 72 h after inoculation, *TaPKS33* and *TaPKS31* were upregulated in the susceptible material 7182, whereas the other six genes were downregulated (*TaPKS68*, *TaPKS71*, *TaPKS72*, *TaPKS66*, *TaPKS70*, and *TaPKS29*). At this time, *TaPKS68* and *TaPKS71* were upregulated in the resistant material H83, whereas the remaining six genes (*TaPKS31*, *TaPKS72*, *TaPKS60*, *TaPKS70*, *TaPKS29*, and *TaPKS33*) were downregulated (Figure 9B).

At the two time points after inoculation, only the expression profile of *TaPKS31* was exactly the same in the two materials, whereby its expression was upregulated at both time points. The expression patterns of *TaPKS33*, *TaPKS68*, and *TaPKS71* were the opposite in the two materials; the expression of *TaPKS33* was upregulated initially and then downregulated in the susceptible material 7182, whereas it was downregulated initially and then upregulated in the resistant material H83. *TaPKS68* and *TaPKS71* were continuously downregulated in the susceptible material but upregulated in the resistant material. Therefore, we suggest that these eight *TaPKS* family genes are involved in the response to wheat sheath-blight and the regulation of disease resistance. The differences in the expression and regulation of some *PKS* genes in the two materials may explain the differences in their resistance to *Rhizoctonia cerealis*.

### 2.8. Expression Profiles of Four TaPKS Genes Analyzed via qRT-PCR

Based on the RNA-seq data, we screened eight *TaPKS* genes with significant differences in expression. In order to further understand the *TaPKS* gene expression patterns in wheat with differences in sheath-blight resistance, qRT-PCR was used to detect the *TaPKS* gene expression levels at 6 h, 12 h, 24 h, 36 h, 72 h, and 7 days after inoculating resistant and susceptible wheat with *Rhizoctonia cerealis* (Figure 10). Specific primers could not be designed for the *TaPKS68*, *TaPKS29*, *TaPKS33*, and *TaPKS68* genes, so they were not analyzed. Finally, we performed qRT-PCR analysis for four *TaPKS* genes. At 36 h after inoculation, the *TaPKS70*, *TaPKS72*, and *TaPKS66* genes were downregulated in both materials compared with their respective controls, whereas *TaPKS31* was upregulated in both materials; however, the times with upregulated expression were much higher in the disease-resistant material than the susceptible material. At 72 h after inoculation, *TaPKS70*, *TaPKS72*, and *TaPKS31* were downregulated in both materials, but *TaPKS66* was upregulated in the resistant material and downregulated in the susceptible material. These results confirm the reliability of the RNA-seq data. In addition, we detected the expression of *TaPKS* genes at other time points after inoculation (6 h, 12 h, 24 h, and 7 days). *TaPKS70* was upregulated at 6 h after inoculation in the resistant material and at 24 h after inoculation in the susceptible material. *TaPKS72* was downregulated in both materials after inoculation. *TaPKS66* was upregulated in the resistant material at 6 h and 7 days after inoculation, but downregulated in the susceptible material at all time points after inoculation (6 h, 12 h, 24 h, and 7 days). *TaPKS31* was upregulated in the resistant material at 12 h and 24 h after inoculation, but downregulated in the susceptible material at all time points (6 h, 12 h, 24 h, and 7 days). Therefore, we suggest that the response times and expression patterns of these *TaPKS* genes in the different materials may explain their differences in resistance.

### 2.9. Response Patterns of Four TaPKS Family Genes after Hormone Treatment

Plant hormones such as salicylic acid, jasmonic acid, abscisic acid, and ethylene are signaling molecules that play important roles in plant resistance to biotic and abiotic stresses. Wheat plants were sprayed with these hormones at the third leaf stage, and the changes in *TaPKS* gene transcription under different hormone treatments were analyzed in order to understand the responses of *TaPKS* genes to different hormones (Figure 11). The results showe that *TaPKS* gene expression was significantly induced by hormone treatment, especially *TaPKS72*. Under salicylic acid treatment, the first peak occurred 1 h after treatment, before decreasing and then increasing again from 3 h to 12 h (Figure 11A). Under treatment with methyl jasmonate and ethylene, the first peak occurred 1 h after treatment, before gradually decreasing (Figure 11B,C). Under abscisic acid treatment, the response of *TaPKS72* was more complex, with two peak values at 1 h and 6 h after treatment (Figure 11D). In addition, *TaPKS70* and *TaPKS66* were significantly induced by the hormones, whereas *TaPKS31* was weakly induced. These results further show that the expression of *TaPKS* genes was regulated by salicylic acid, methyl jasmonate, ethylene, and abscisic acid. The response of *TaPKS* genes to hormonal induction may be related to the presence of hormonal response elements in their promoter. Therefore, we speculate that *TaPKS* genes may participate in the process of wheat sheath-blight resistance by mediating hormonal changes in response to *Rhizoctonia cerealis* infection.

## 3. Discussion

Plants are often attacked by various pathogens, and they have evolved various defense mechanisms. In particular, previous studies have focused on plant disease resistance mediated by the PR gene, NBS-LRR gene, and Ser/Thr protein kinase [23,24,25]. In recent years, great progress has been made in understanding plant disease resistance, particularly the plant disease resistance signal-transduction pathway; however, some issues still need to be clarified. When a host plant is infected by a pathogen, various mechanisms can help it to resist the infection. In particular, plants have specialized structures such as stomata, thickened cell walls, and callose, and they can secrete antibacterial substances [26,27]. Moreover, plants can resist infection by pathogens via responses such as hypersensitivity, programmed cell death, and hormone-induced resistance. Any link in this regulatory network is indispensable. Plant *PKS* genes belong to a polygenic family [28] with important roles in physiological activities, such as plant anthocyanins [29], plant fertility [30], and plant defense responses [29,31].

### 3.1. PKS Family in Plants

In plants, CHS is the key rate-limiting enzyme in polyketide metabolism, whereby it catalyzes the production of flavonoids to participate in plant development and resist infection by pathogens [32]. The functions and regulation of *PKS* family genes have been investigated in many plants. For example, 5 *PKS* genes were identified in tobacco [33], 8 in *Petunia hybrida* [34], 6 in common morning glory (*Ipomea purpurea*) [35], 9 in flax (*Linum usitatissimum* L.) [36], 3 in *Aquilaria sinensis calli* [37], 14 in soybean (*Glycine max*) and corn *(Zea mays* L.) [38,39], 8 in pea [40], and 27 in rice [41,42]. In the present study, we identified 81 *TaPKS* genes with complete domains, which is three times the number of *PKS* genes present in rice. This difference may have been caused by genomic duplication events during wheat evolution. According to the *PKS* genes previously identified in other species, *TaPKS* genes can be divided into two categories (Figure 2), which is consistent with the classification in rice [41].

### 3.2. Intraspecific and Interspecific Relationships of TaPKS Genes

Tandem duplication and segmental duplication are the main routes for gene family expansion in the genome evolution process. Studies have shown that gene duplication has occurred in 70–80% of angiosperms [43]. Gene duplication is the basis of species evolution. Our analysis of gene duplication events showed that the 81 *TaPKS* genes contain 8 pairs of tandem duplicated genes and 33 pairs of segmental duplicated genes, thereby indicating that tandem duplication and segmental duplication events accelerated the expansion of the *TaPKS* gene family. We analyzed the homologous relationships between *PKS* genes in wheat and other species, and identified 49 pairs of syntenic *PKS* genes in wheat and *T. dicoccoides*, 4 pairs in wheat and maize, and 3 pairs in wheat and rice. Thus, more homologous events were detected between wheat and *T. dicoccoides*, which is consistent with the closer evolutionary distance between wheat and *T. dicoccoides*. In addition, the Ka/Ks values calculated for *TaPKS* genes in wheat were less than 1, and thus, this gene family has experienced strong purifying selection in the evolutionary process. This purification selection has eliminated harmful mutations at duplicated loci to maintain the important biological functions of the members of this gene family.

### 3.3. Structures and Conserved Motifs in TaPKS Genes in Wheat

The structures of genes affect their functions. The evolution of gene families is accompanied by the gain and loss of introns [44]. Introns function by regulating gene expression, and they can increase transcription levels by affecting the transcription rate, nuclear output, and stability of transcription [45]. Introns can also improve the efficiency of mRNA transcription [46]. In the present study, we showed that most of the 81 *TaPKS* have one–three exons and zero–three introns. In particular, 47 *TaPKS* with one intron and two exons comprise the highest proportion, whereby they account for 58% of the total *TaPKS* genes, which is consistent with a previous report stating that most *CHS* genes contain one intron and two exons [47]. Some wheat *PKS* family genes also have no introns and multiple introns, as found in other plants, but the specific mechanism involved is still unclear [47,48,49,50]. Ten conserved motifs are found in the *TaPKS* family, wherein motif 1, motif 3, motif 5, motif 7, and motif 10 are conserved in all members. However, some genes also have unique motifs, which can be used as the basis for gene classification and functional differentiation.

### 3.4. Cis-Acting Elements and Hormone Response Analysis

The expression of genes is mostly determined by upstream promoters. Specific combinations of cis-acting elements and transcription factors in promoters are most important for biological signal transduction, and they are also important for synergy with other genes [51]. Many cis-acting elements related to disease resistance have been found in the promoters of plant pathogenesis-related genes, such as a the GCC-box element that specifically binds to ERF transcription factors [52], and W-box elements that bind to WRKY transcription factors [53]. In addition, many elements related to hormone responses have been identified [54]. In the present study, we found that all 81 *TaPKS* genes contain hormone-response-related elements, whereby methyl jasmonate and abscisic acid responsive elements comprise the highest proportion. Thus, we suggest that the expression of *TaPKS* genes may be regulated by hormones. Previous studies have shown that the expression of *CHS* genes in soybean, parsley, and *Picea glauca* is induced by JA and its esters, including methyl jasmonate [55,56,57], and the expression of *PaCHS* in *Plagiochasma appendiculatum* is increased via treatment with methyl jasmonate, abscisic acid, or salicylic acid [1]. In addition, *AsCHS* is regulated by methyl jasmonate, abscisic acid, and salicylic acid in *Aquilaria sinensis* [37]. We found that the expression levels of *TaPKS* genes were induced and regulated by hormones comprising ethylene, methyl jasmonate, salicylic acid, and abscisic acid, and the *TaPKS* genes responded differently to each hormone. Similarly, differences have previously been found in the expression of *PKS* genes under induction by methyl jasmonate in wheat, soybean, parsley, spruce, and *Plagiochasma appendiculatum*. In conclusion, we suggest that *PKS* genes may be necessary to protect plants via hormone-dependent signaling pathways.

### 3.5. Transcriptome Analysis Combined with qRT-PCR to Determine the Expression Profiles of TaPKS Family Members

Each type-III *PKS* can be regulated in different ways, such as tissue-specific expression, transcription induced by light treatment or pathogen infection, and transcription controlled by other regulatory genes. For example, *CHS* genes are mainly expressed in flower tissues in mature plants [58]. The expression of the stilbene synthase gene is mainly concentrated in the fruit in grape and peanut, but in the root tissue in *Polygonum cuspidatum* and rhubarb [9,16]. In the present study, RNA-seq data showed that most *TaPKS* genes were specifically expressed in the anthers and panicles, which is consistent with a previous report in which *PKS* was mainly expressed in the flowers in gentian (*Gentiana triflora*) [58]; thus, we suggest that *TaPKS* may be related to anther development [47]. In addition, type-III polyketide is part of the plant defense mechanism, so many plants express different types of type-III *PKS* genes when stimulated by stresses such as pathogen infection, ultraviolet radiation, ozone, and trauma [59,60]. In this study, many *TaPKS* genes were induced via infection with *Rhizoctonia cerealis*. Therefore, we suggest that *TaPKS* may play a role in wheat sheath-blight resistance by regulating the synthesis of flavonoids. Further research is needed to understand how *TaPKS* genes play roles in plant disease resistance by regulating the synthesis of polyketones.

## 4. Materials and Methods

### 4.1. Material Planting and Treatments

Sheath blight-resistant wheat material H83 and moderately susceptible wheat material 7182 were used as test materials, which were cultured in an artificial climate room. First, the seeds were germinated in Petri dishes covered with absorbent paper, before transferring them to pots containing soil for vernalization treatment in a vernalization incubator for 30 days. Finally, the plants were moved into an artificial climate chamber until the jointing stage. The day/night temperatures and photoperiod settings in the artificial climate chamber were 18 °C/16 °C and 14 h/10 h, respectively, and the humidity was set to 70%. Sheath blight strain R0301 used in this study was provided by Jiangsu Academy of Agricultural Sciences. The toothpick method was used to inoculate the wheat material at the jointing stage. A toothpick covered with mycelium was placed in the gap between the sheath and stem of the penultimate leaf at the base of a wheat plant, before wrapping with sterilized wet absorbent cotton. Only the main stem of each wheat plant was inoculated. The day/night temperatures in the artificial climate chamber after inoculation were adjusted to 24 °C/20 °C and the humidity was 80%. R0301 was grown on potato dextrose agar medium covered with radial toothpicks for about two weeks before inoculation, and the toothpicks were covered with the white mycelium of *Rhizoctonia cerealis*. The penultimate leaves at the base were collected at 0 h, 36 h, and 72 h after inoculation, with three biological replicates at each time point. 

In the third leaf stage, the resistant material H83 was treated with 1 mM salicylic acid, 0.1 mM ethylene, 0.2 mM abscisic acid, or 0.1 mM methyl jasmonate, and the control was treated with 0.1% Tween-20. This treatment referred to previous studies [61]. Leaf samples were collected after 0 h, 1 h, 6 h, and 12 h. The collected leaf samples were rinsed with distilled water and then placed in a 5 mL cryopreservation tube before rapid freezing in liquid nitrogen, and then stored at –80 °C. All of the materials used in this study were planted in parallel. 

### 4.2. Identification of PKS Family Members in Wheat Genome

Pfam PF00195 and PF02797, corresponding to conserved protein domains Chal_sti_synt_N and Chal_sti_synt_C, respectively, were used to extract wheat *PKS* family members from the Ensembl Plants database [62]. Protein sequences without a complete PKS domain (Chal_sti_synt_N and Chal_sti_synt_C) were removed after their identification in NCBI-CDD, and redundant transcripts were also removed [63].

### 4.3. Physicochemical Properties, Chromosomal Localizations, and Gene Duplication in TaPKS Family Genes

The ProtParam tool in ExPASY [64] was used to predict the number of amino acids, molecular weight, isoelectric point, and grand average of hydropathicity (GRAVY) for *TaPKS* family genes. *TaPKS* family genes were named according to the order of their positions on chromosomes. Chromosome location information for *TaPKS* family genes was determined using the Ensembl Plants database and via mapping with TBtools software [65]. MCScanX was used to analyze gene replication events in the wheat genome and homology between *PKS* family genes in wheat and other selected species [66]. The Ka/Ks ratio was calculated using the online tools MEGA7.0 and DnaSP 6.

### 4.4. Gene Structure, Conserved Domains, and Phylogenetic Analysis of TaPKS Family Genes

The GFF3 file containing the *TaPKS* gene structures in wheat was downloaded from Ensembl Plants, and gene structures were drawn using TBtools. MEME tool was used to analyze the motifs in wheat *TaPKS* protein sequences, whereby the motif length was set to 10–100 and the number of motifs was set to 10 [67]. The *PKS* gene family protein sequences in *Arabidopsis*, rice, barley, maize, alfalfa, woodland tobacco, white campion, *Phalaenopsis* sp., and *Bromheadia finlaysoniana* were downloaded from UniProt. Multiple-sequence alignment was conducted for PKS proteins with Jalview to analyze their evolutionary relationships. The phylogenetic tree was constructed using the neighbor-joining method in MEGA11 software with 1000 bootstrap tests. The evolutionary tree was edited and refined using FigTree v1.4.4.

### 4.5. Analysis of Cis-Acting Elements in PKS Gene Promoter in Wheat

PlantCARE was used to predict cis-acting elements in the 2000 bp upstream region of the gene initiation codon [68] and visualized using TBtools.

### 4.6. qRT-PCR Analysis of TaPKS Gene Expression Levels

Total RNA was extracted from the plant materials using an RNA extraction kit supplied by Tiangen Biotech (Beijing) Co. Ltd., cDNA was then synthesized using an RNA reverse transcription kit. The specific primers were designed using Oligo 7 software and submitted to Beijing Qingke Biotechnology Co. Ltd. for synthesis. We quantified the gene expression levels by using actin as an internal reference gene. qRT-PCR was performed using the Applied Biosystems 7500 Real-Time PCR system. The 20 μL reaction mixture contained 10 μL 2× SuperReal PreMix Plus, 0.6 μL forward primer, 0.6 μL reverse primer, 1.8 μL cDNA, 0.4 μL 50×ROX Reference Dye, and 6.6 μL double-distilled H_2_O. The PCR cycle conditions comprised 15 min at 95 °C, followed by 40 cycles at 95 °C for 10 s and 60 °C for 32 s. Data were expressed as the mean ± standard error based on three independent replicates. Relative quantification levels of *TaPKS* genes were calculated using the 2^−ΔΔCT^ method. The primers used for qRT-PCR are listed in Appendix A.

### 4.7. RNA-Seq Data Analysis

Wheat transcriptome data for different tissues were downloaded from the Plant Public RNA-seq database [22]. RNA-seq data were obtained for five different tissues in Chinese Spring Wheat under normal growth conditions. In order to study the regulatory mechanism associated with *TaPKS* genes in wheat infected with sheath blight, we analyzed transcriptome data for resistant and susceptible wheat infected with *Rhizoctonia cerealis* at different time points [69]. The transcriptome data are available from the following website: https://www.ncbi.nlm.nih.gov/sra/PRJNA749387 (accessed on 1 December 2021). Genes with an adjusted *p*-value < 0.01 and|log_2_foldchange (FC)|> 1 according to DESeq2 were designated as differentially expressed genes. A heat map was prepared using the OmicShare tools.

## 5. Conclusions

In this study, we systematically identified members of the type-III *PKS* gene family in the wheat genome for the first time. The family contains 81 members. Similar to those found in other species, the *PKS* genes in wheat can be divided into two categories comprising *CHS* and non-CHS genes. Expansion of the wheat *PKS* gene family mainly occurred via tandem duplication and segmental duplication events. Synteny analysis showed that the *PKS* genes in wheat and *T. dicoccoides* are closely related. In addition, by predicting the promoter elements of the *TaPKS* gene, it was found that the promoter region of the *TaPKS* gene contains many hormone regulatory elements. Therefore, we speculate that *TaPKS* genes may be regulated by hormones. After that, we analyzed the expression pattern of *TaPKS* genes in the resistant material H83 treated with salicylic acid, methyl-jasmonate, abscisic acid, and ethylene via qRT-PCR; this verified the hypothesis that the expression of *TaPKS* genes is regulated by hormones (salicylic acid, methyl-jasmonate, abscisic acid, and ethylene), indicating that *TaPKS* genes may participate in the process of wheat sheath-blight resistance by mediating hormonal changes in response to *Rhizoctonia cerealis* infection. Previous RNA-seq data showed that the expression of *TaPKS* genes was tissue-specific and induced by *Rhizoctonia cerealis*. The results of qRT-PCR confirmed that the expression of *TaPKS* was induced by *Rhizoctonia cerealis*. In addition, we believe that that the response times and expression patterns of these *TaPKS* genes in the different materials were related to the resistance of these materials to wheat sheath-blight. In conclusion, wheat *PKS* genes play important roles in plant resistance to pathogens. Our results provide an important and comprehensive reference framework for further study of the wheat *PKS* gene family, and a useful basis for screening candidate *Rhizoctonia cerealis*-resistance genes.

## Figures and Tables

**Figure 1 ijms-23-07187-f001:**
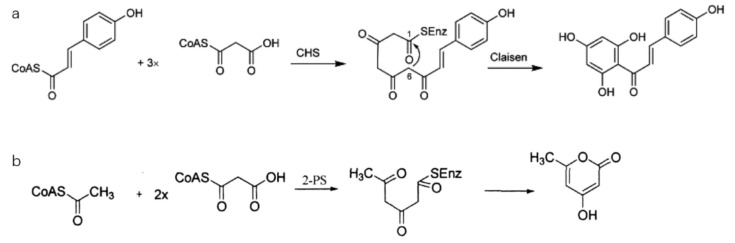
Reaction catalyzed by type-III PKSs.

**Figure 2 ijms-23-07187-f002:**
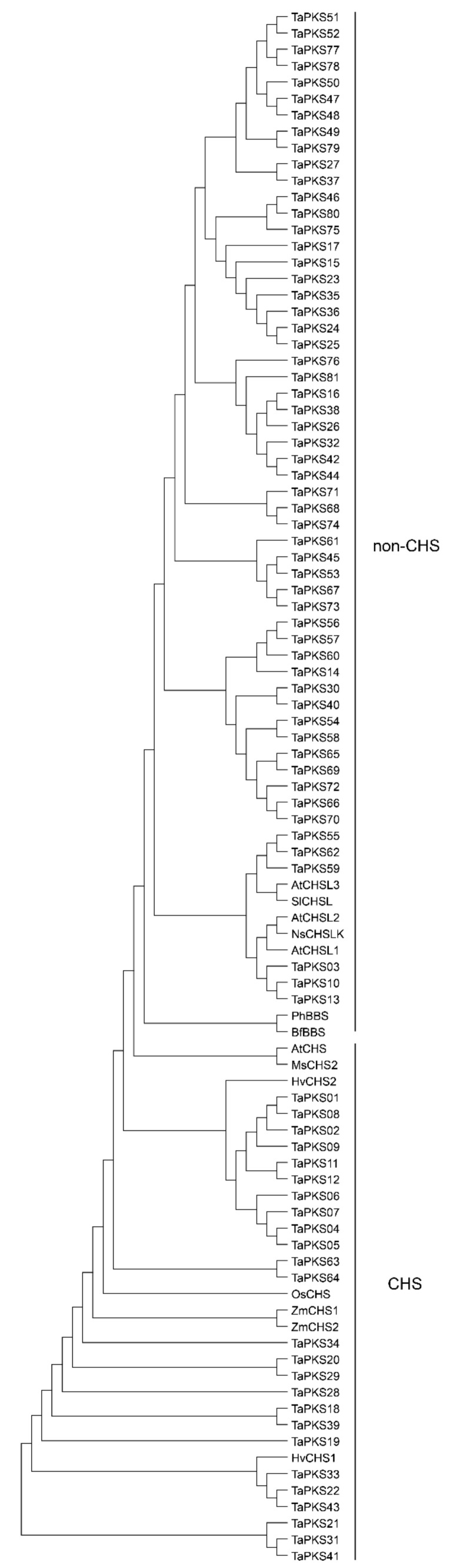
Evolutionary tree constructed based on *PKS* family genes in wheat and other species. The CHS superfamily includes CHS and a series of chalcone synthase-like (CHSL, also called non-CHS) genes [20,21]. Their functional differences are mainly reflected in the selection of initial substrates and the number of condensation reactions (polyketide chain extension length) [8,9,10].

**Figure 3 ijms-23-07187-f003:**
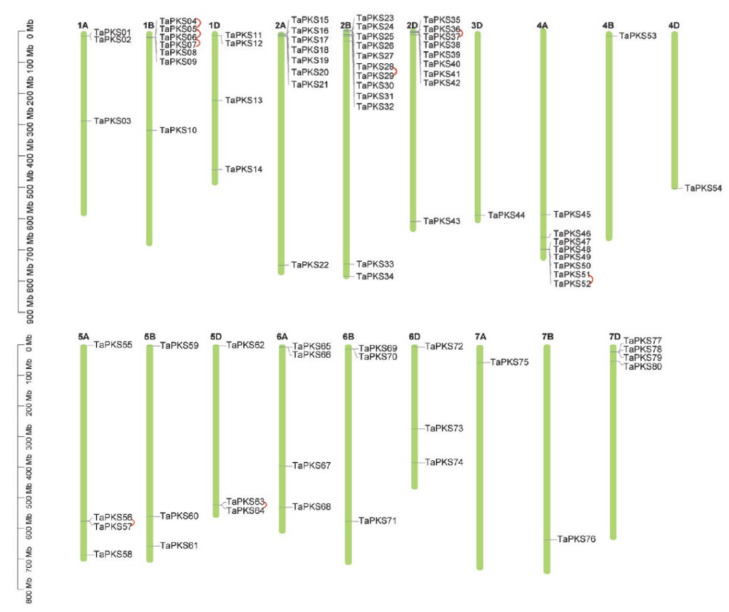
Chromosome locations of *PKS* family genes in wheat. Black font indicates tandem duplication genes, which are connected by red arcs. The scale on the left indicates the length of the chromosome in millions of bases (Mb).

**Figure 4 ijms-23-07187-f004:**
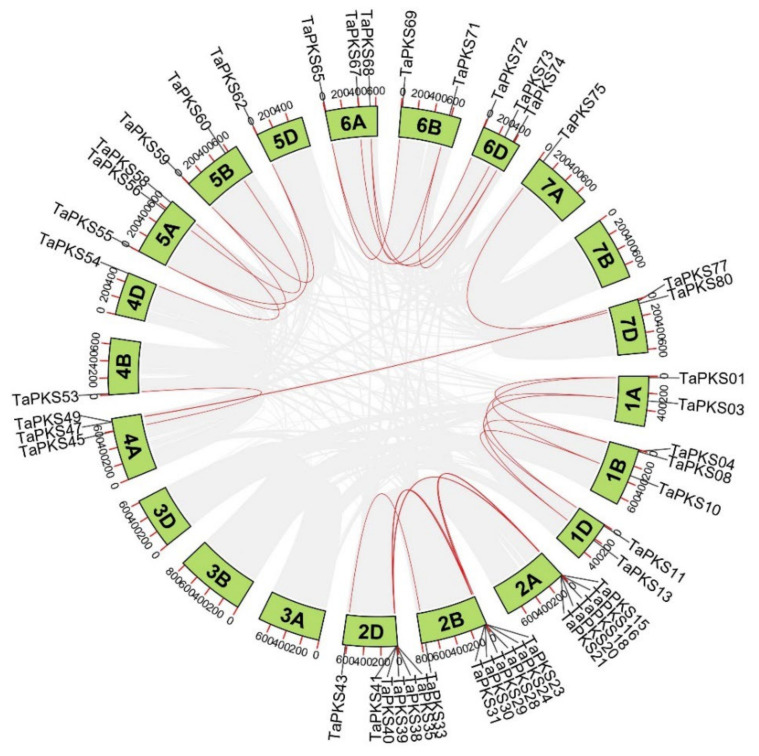
Chromosomal distribution of *PKS* segmental duplication genes in wheat. The gray lines in the background represent the segmental duplication in the whole wheat genome. Red lines represent *PKS* gene pairs with segmental duplication.

**Figure 5 ijms-23-07187-f005:**
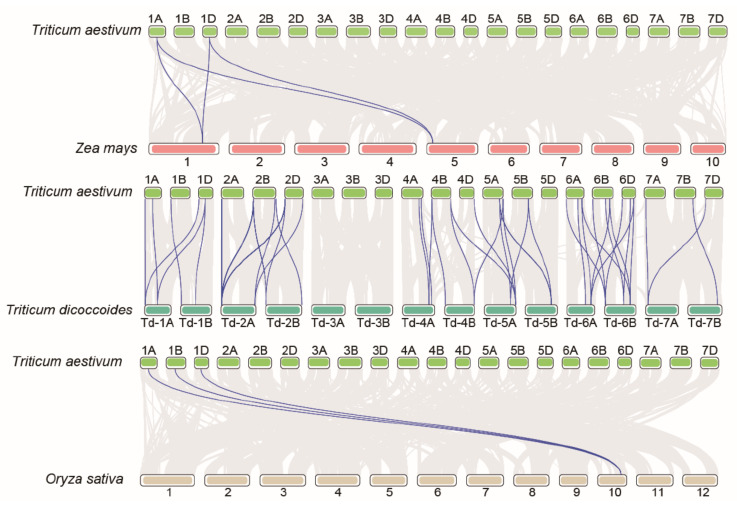
Synteny analysis for *PKS* genes in wheat and representative species comprising *T. dicoccoides*, maize, and rice. The gray lines in the background represent the synteny regions in wheat and the genomes of other species, and the blue lines represent the *PKS* gene pairs with synteny in wheat and other species.

**Figure 6 ijms-23-07187-f006:**
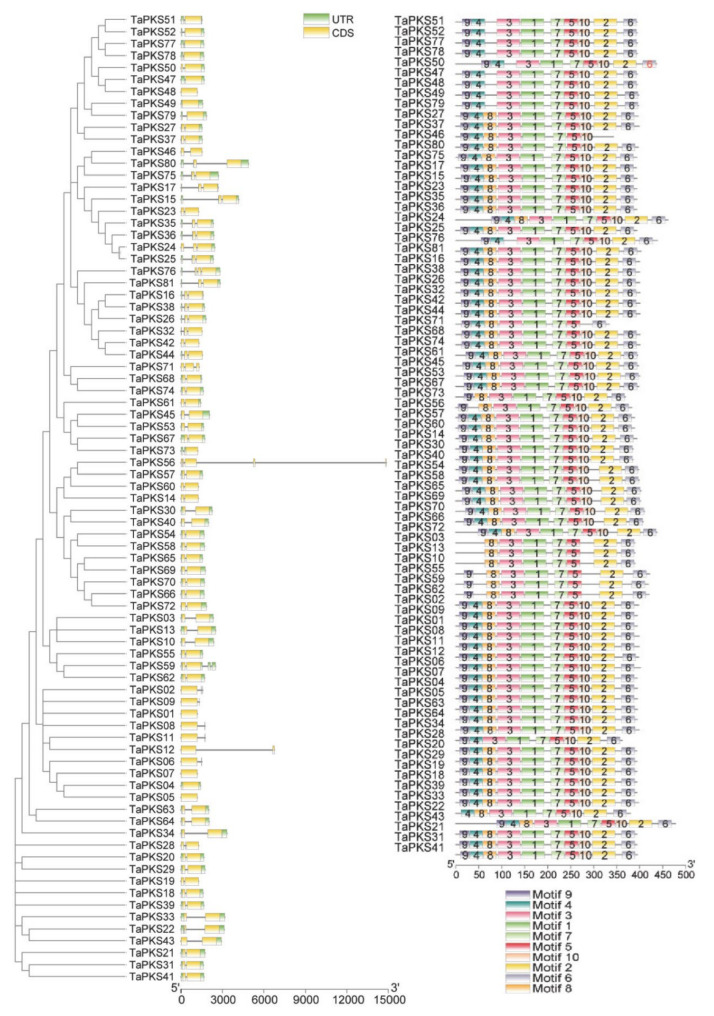
Gene structure and conserved domains in *TaPKS* family genes. The structural composition of the gene is shown on the left. The yellow boxes represent exons, the green boxes represent untranslated regions, and the gray lines connecting the two colored boxes represent introns. The distribution of conserved domains is shown on the right and the boxes with different colors represent different motifs. The scale bar at the bottom of the figure shows the gene length in base pairs.

**Figure 7 ijms-23-07187-f007:**
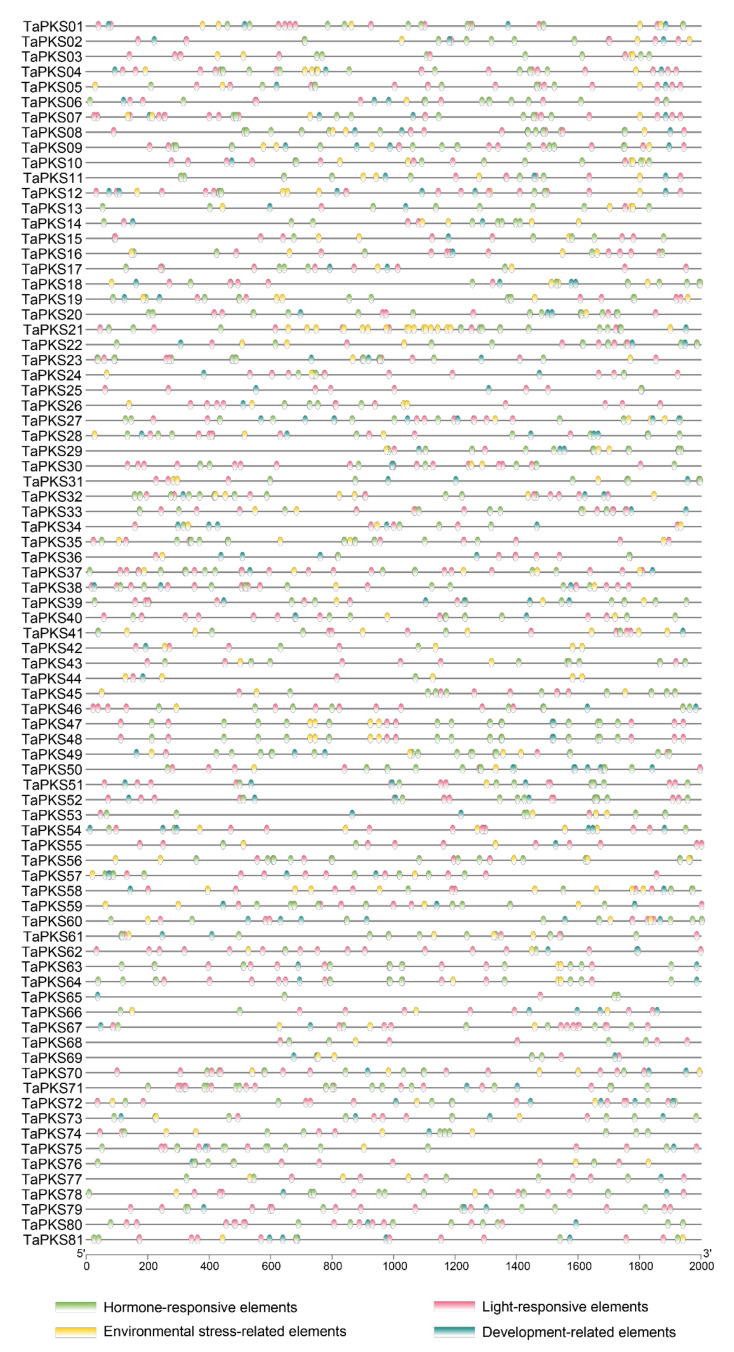
Predicted cis-acting elements in promoters of *PKS* genes in wheat. The ellipses with different colors represent different promoter elements and the scale at the bottom shows the length of the promoter.

**Figure 8 ijms-23-07187-f008:**
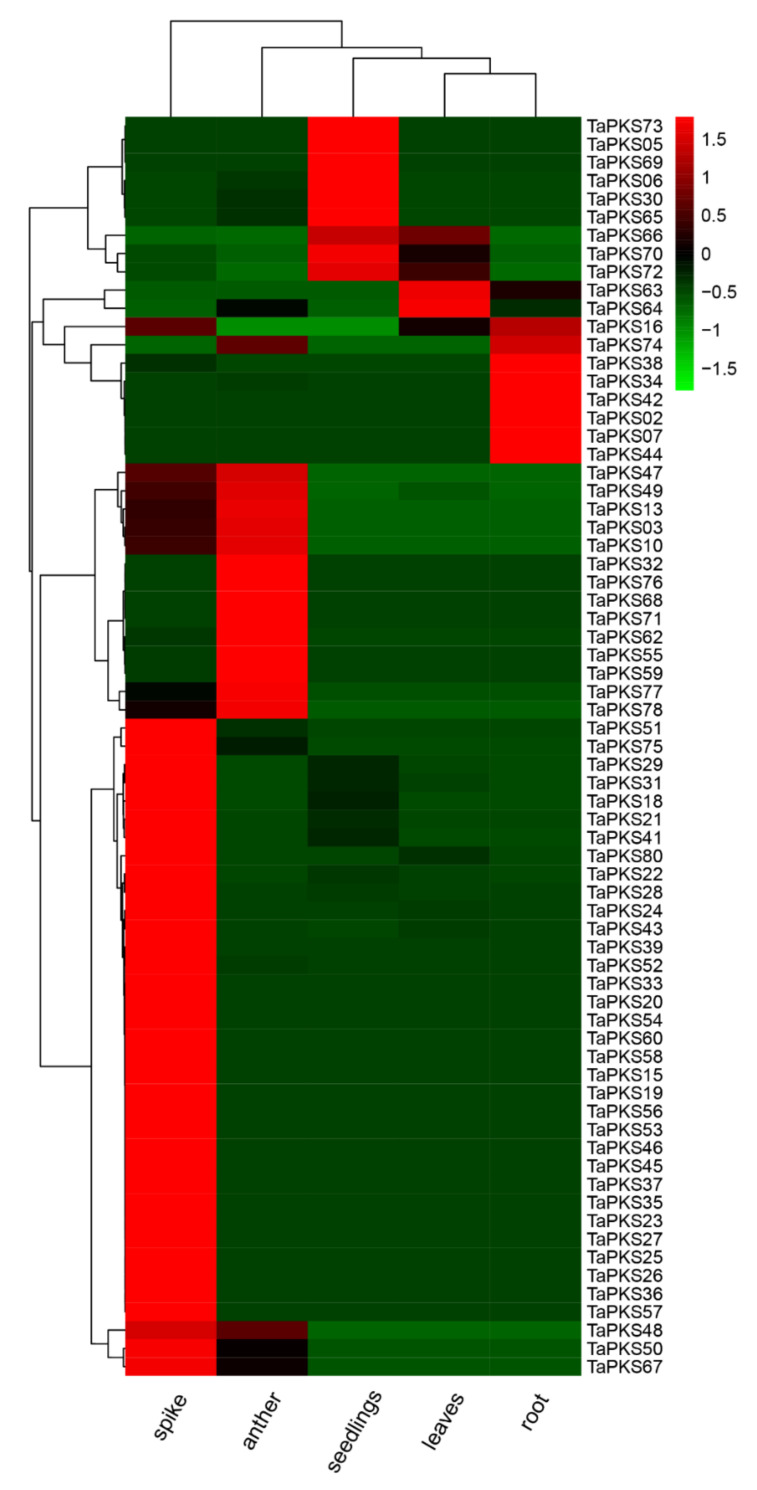
Cluster analysis of *TaPKS* gene expression in different tissues from Chinese Spring wheat. The tissue expression data used for mapping were obtained from Plant Public RNA-seq Database [22]. The change in the color of the box from green to red indicates that the gene expression level changes from low to high.

**Figure 9 ijms-23-07187-f009:**
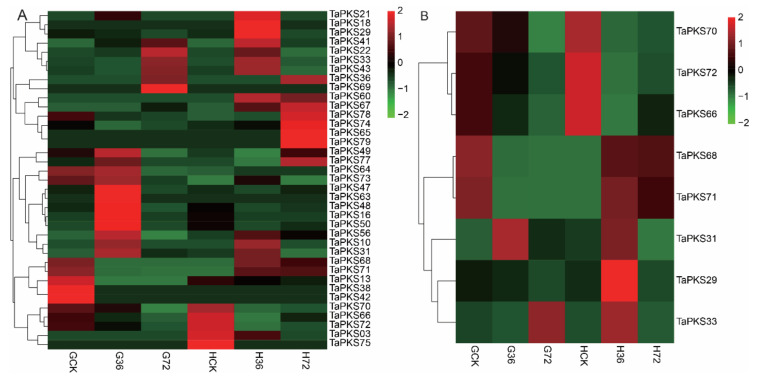
Expression profiles of *TaPKS* genes after infection by *Rhizoctonia cerealis*: (**A**) expression profiles of *TaPKS* genes in the resistant material H83 and susceptible material 7182 after inoculation; (**B**) expression profiles of eight differentially expressed *TaPKS* genes in the resistant material H83 and susceptible material 7182 after inoculation. Values of fragments per kilobase of transcripts per million mapped reads (FPKMs) are represented by a color gradient from low (green) to high (red).

**Figure 10 ijms-23-07187-f010:**
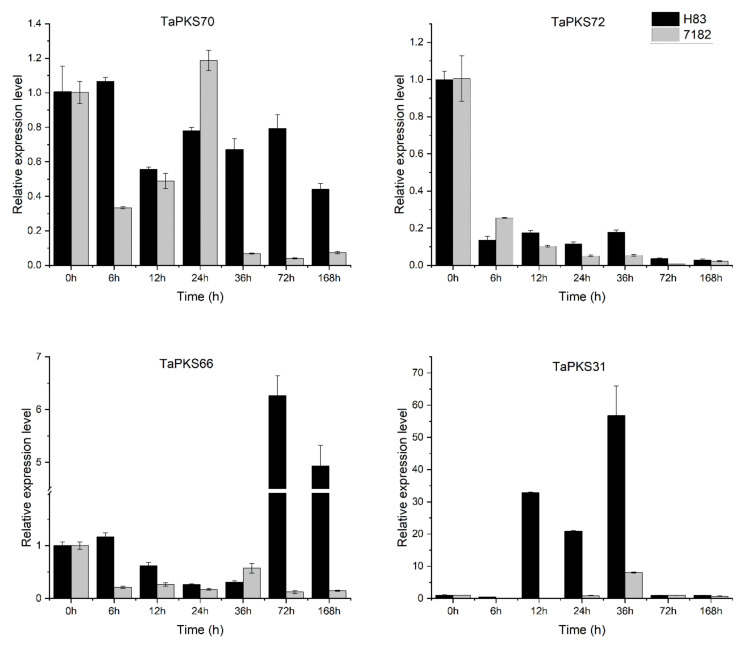
Expression analysis for four *TaPKS* genes in resistant wheat (H83) and susceptible wheat (7182) at different times after inoculation with *Rhizoctonia cerealis*. The average expression values were calculated based on three independent replicates. The error bars on the columns represent standard deviations.

**Figure 11 ijms-23-07187-f011:**
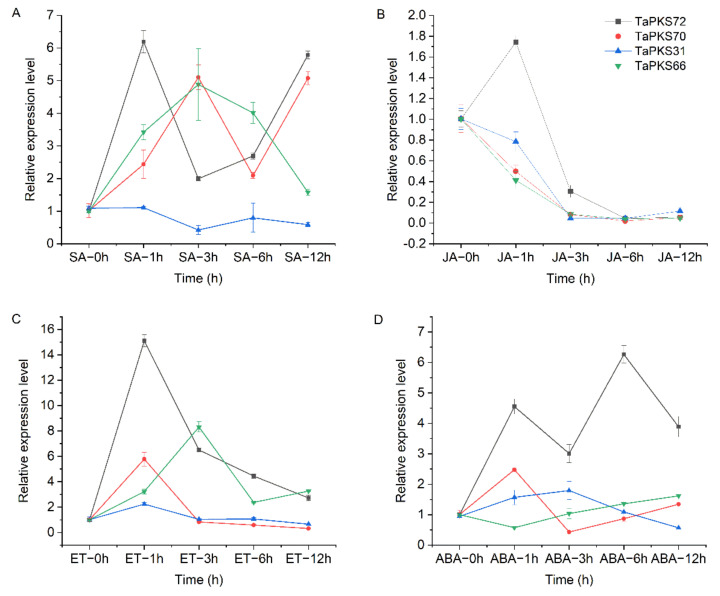
Expression patterns of four *TaPKS* genes in resistant materials after treatment with salicylic acid (SA), methyl jasmonate (MeJA), ethylene (ET), and abscisic acid (ABA). **A**–**D** represent the expression trend of four *TaPKS* genes treated with salicylic acid, jasmonic acid, ethylene and abscisic acid respectively. The average expression values were calculated based on three independent replicates. Error bars represent standard errors.

## Data Availability

Not applicable.

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
