# Peer review of "Genome-Wide Analysis of Type-III Polyketide Synthases in Wheat and Possible Roles in Wheat Sheath-Blight Resistance"

_ijms, 2022, doi:10.3390/ijms23137187_

Round 1

Reviewer 1 Report

The authors need to rewrite their conclusion so that it matches what they suggest this article will do in the abstract and introduction. As it stands it is only half way there.

The same central items need to be address and repeat throughout the manuscript:

• Several (many) of the figures have font sizes that are far too small some can not be read at all.  All text in the figures should be Arial 9 or greater

• Please provide publication references not websites for your software applications. Websites change too often.

• A general read of the manuscript for typographical errors is needed and this reviewer has attached some tracked revisions in the attached PDF file

Author Response

Dear editor/reviewer,

Thank you for your comments and suggestions on our manuscript entitled " Genome-wide analysis of type III polyketide synthases in wheat and possible roles in wheat sheath blight resistance ".

. All the comments are valuable and very helpful for revising and improving our paper, as well as the important guiding significance to our researches. We must thank you for taking the trouble to edit the manuscript. We have studied comments carefully and have made correction which we hope meet with approval. Revised portion are marked up using the“Track Changes” function. The main corrections in the paper and the responds to the comments are as flowing. Should you have any questions, please contact us without hesitate.

Point 1: Line 27: All species names including those in the abstract and main manuscript need to be in italics

Response 1: Thank you for your reminding. After careful examination, we have modified the names of all species involved in this manuscript in italics.

Point 2: Line 30 : no need to say this delete it

Response 2: Thank you for your suggestion. We have deleted it in Line 30.

Point 3: Line 33:Polyketide-derived compounds or just Polyketides. Few of these compounds contain ketones

Response 3:Thank you for your suggestion. According to the reviewer's suggestions, we have modified "polyketone compounds" to "polyketid derived compounds" in line 37.

Point 4: Line 41: delete”only”

Response 4: Thank you for your suggestion. According to the reviewer's suggestions, We have deleted it in Line 41.

Point 5: Line 51: delete” are generated” This sentence should be revised gramatically.

Response 5: Thank you for your suggestion. According to the reviewer's suggestions,

We have deleted "are generated" from the original sentence, and the original sentence has been changed from " In particular, CHS catalyzes the condensation of p-coumaroyl-CoA and three molecules of malonyl-CoA are generated through three enzymatic reactions to produce enzyme-bound tetrakis ketone compound intermediates, before undergoing Claisen-type cyclization to form naringenin chalcone, which is an important intermediate in the biosynthesis of plant flavonoids and isoflavones" to "In particular, CHS catalyzes the condensation of p-coumaroyl-CoA and three molecules of malonyl-CoA through three enzymatic reactions to produce enzyme-bound tetrakis ketone compound intermediates, before undergoing Claisen-type cyclization to form naringenin chalcone (Figure 1a), which is an important intermediate in the biosynthesis of plant flavonoids and isoflavones"

Point 6: Line 53: A figure showing this here or in the SI would help

Response 6:Thank you for your suggestion. According to the reviewer's suggestions, we present the catalytic process in a graph, and the specific catalytic process is shown in the figure 1a below.

Figure 1. Reaction catalyzed by type III PKSs.

Point 7: Line 55: If a figure is made please include this as well

Response 7: According to the reviewer's suggestions, we present the catalytic process in a graph, and the specific catalytic process is shown in the figure 1b above.

Point 8: Line 82: The abbreviations here are not common as most plant biologists simply write the names out. I understand their use in figures but suggest that these abbreviations be dropped to help the usual reader digest the manuscript.

Response 8: Thank you for your suggestion. According to the reviewer's suggestions, we have deleted these abbreviations.

Point 9: Line 105: It would help if the caption defined CHS and non-CHS for the reader who only looks at the figures.

Response 9: Thank you for your suggestion. According to the reviewer's suggestions, we have added the definitions of CHS and non CHS in the caption of Figure 2, specifically described as” The CHS superfamily includes CHS and a series of chalcone synthase-like (CHSL, also called non-CHS) genes [20-21]. Their functional differences are mainly reflected in the selection of initial substrates and the number of condensation reactions (polyketide chain extension length [8-10].” in Line 110-112.

  1. Beerhues, L.; Liu, B., Biosynthesis of biphenyls and benzophenones – Evolution of benzoic acid-specific type III polyketide synthases in plants. Phytochemistry 2009, 70, (15), 1719-1727.
  2. Katsuyama, Y.; Kita, T.; Funa, N.; Horinouchi, S., Curcuminoid biosynthesis by two type III polyketide synthases in the herb Curcuma longa. The Journal of biological chemistry 2009, 284, (17), 11160-11170.
  3. Mizuuchi, Y.; Shi, S.-P.; Wanibuchi, K.; Kojima, A.; Morita, H.; Noguchi, H.; Abe, I., Novel type III polyketide synthases from Aloe arborescens. The FEBS Journal 2009, 276, (8), 2391-2401.
  4. Han, Y.; Zhao, W.; Wang, Z.; Zhu, J.; Liu, Q., Molecular evolution and sequence divergence of plant chalcone synthase and chalcone synthase-Like genes. Genetica 2014, 142, (3), 215-225.
  5. Yang, J.; Gu, H., Duplication and divergent evolution of the CHS and CHS-like genes in the chalcone synthase (CHS) superfamily. Chin. Sci. Bull 2006, 51, (5), 505-509.

Point 10: Line 126: Please increase the font size in this figure. There is plenty of white space left and one can easily see Arial 9 or the equivalent fitting.

Response 10: Thank you for your suggestion. We have increased the font in the figure (Figure 3) in Line 133.

Point 11: Line 137: One typically abbreviates this as T. dicoccoides after the first use repeat throughout the entire manuscript for a species names.

Response 11: Thank you for your reminding. In the whole manuscript, except for the first use of the species name Triticum dicoccoides, all the species names (Triticum dicoccoides) are abbreviated as T.dicoccoides.

Point 12: Line 145: Species names should be put in non bold italics and remove the _

These are computer based names, while ok for the computer they are not ok for the publication

Also please make sure that all fonts are at least 9 in size fonts 1A... and 1,2,3... are far too small.

Response 12: Thank you for your suggestion. We have adjusted the font and remove the” _” in Line 151.

Point 13: Line 166: The fonts are too small. ABSOLUTELY nothing can be read in this figure. All fonts need to be at least Arial 9 or greater. This must be redone.

Response 13: Thank you for your suggestion. We have increased the font size in Line 172.

Point 14: Line 194: Remove the bolding of the text in this figure it is not needed. All fonts should be the same size? Arial 10 would fit.

Response 14: Thank you for your suggestion. We have increased the font size in Line 200.

Point 15: Line 202: Modify "42, 16, eight, nine, and four" to "42, 16, 8, 9, and 4"

Response 15: Thank you for your suggestion. We have modified "42, 16, eight, nine, and four" to "42, 16, 8, 9, and 4" in Line 209.

Point 16: Line 211: please provide the reference here or data set name or code so the reader can find it if needed.

Response 16: Thank you for your suggestion. We have provided the reference in Line 219, reference No. [22].

  1. Yu, Y.; Zhang, H.; Long, Y.; Shu, Y.; Zhai, J., Plant Public RNA-seq Database: a comprehensive online database for expression analysis of ~45 000 plant public RNA-Seq libraries. Plant Biotechnol. J 2022, 20, (5), 806-808.

Point 17: Line 220: I don't believe the Journal's format is Figure 8A. Please check and use the correct format.

Response 17: Thank you for your suggestion. We adjusted the format of Figure 8A in Line 260.

Point 18: Line 247: Again the fonts are too small making this figure impossible to read

Response 18: Thank you for your suggestion. We have increased the font size in Line 260.

Point 19: Line 251: Please provide levels for low and hi

Response 19: Thank you for your suggestion. To determine whether the expression levels of a gene under different treatments are increased or decreased, we compared the expression level of the gene under different treatments with the expression level of the gene in the corresponding control. If the expression value is higher than the value of the control group, we believe that the expression level is increased, and if the expression value is lower than the value of the control group, we believe that the expression amount is decreased. The color in the figure was presented according to the relative expression levels of the gene in different materials and treatments. Different genes may show the same color in the figure, but the expression levels represented by the color may be different. That is, the expression level of a gene under this treatment is green, which only represents that the expression level of the gene is relatively low compared with that under other treatments. Similarly, red represents that the expression level of the gene is relatively high compared with that under other treatments. So we can 't define the level of expression with an exact value.

Point 20: Line 277: please convert 7d into hours to fit with the x-axis on each of these plots

Response 20: Thank you for your suggestion. We have converted 7d to hours in Line 290.

Point 21: Line 278: Italics

Response 21: Thank you for your reminding. We have adjusted the font to Italic in Line 291.

Point 22: Line 282: again drop the abbreviations

Response 22: Thank you for your suggestion. We have deleted the abbreviation

Point 23: Line 294: which ones and write 2-3 sentences on what this could mean in terms of its regulation and hormone pathway

Response 23: Thank you for your suggestion. We have modified the description of this section by changing the original sentence " These results further suggest that the expression of TaPKS genes is regulated by hormones, thereby indicating that TaPKS genes function in multiple phytohormone signal transduction pathways." to " These results further showed that the expression of TaPKS genes was regulated by salicylic acid, methyl jasmonate, ethylene and abscisic acid. The response of TaPKS genes to hormonal induction may be related to the presence of hormonal response elements in their promoter. Therefore, we speculate that TaPKS genes may participate in the process of wheat sheath blight resistance by mediating hormonal changes in response to Rhizoctonia cerealis infection.” in Line 310-315.

Point 24: Line 298: Fonts are too small

Response 24: Thank you for your suggestion. We have increased the font size in Line 317.

Point 25: Line 299: You have two sets of abbreviations for each SA and A remove one. I really do not think you need the SA, MeJA and ET.

Response 25: Thank you for your suggestion. According to your suggestion, we have changed the abbreviation SA, MeJA and ET to the full name.

Point 26: Line 419: Add a reason as to why these concentrations were selected.

Response 26: Thank you for your suggestion. We set the hormone concentration based on previous similar studies, and we have added the reference, reference No. [61].

  1. Zhu, X. Functional and Mechanism Analysis of Wheat Genes TaRCR1 and TaAGC1 in Defense Response to Sharp Eyespot. Doctor, Chinese Academy of Agricultural Sciences Beijing, 2016.

Point 27: Line 423: delete not needed

Response 27: Thank you for your suggestion. We have deleted it in Line 447.

Point 28: Line 428: Put in references such as DOI: 10.1007/978-1-4939-6658-5_1

Response 28: Thank you for your suggestion. We have deleted the website and quoted the references you kindly provided in Line 451.

Point 29: Line 430: Put in a real reference and remove the website DOI: 10.1093/nar/gkw1129

Response 29: Thank you for your suggestion. We have deleted the website and quoted the references you kindly provided in Line 455.

Point 30: Line 435: Put in reference and remove website DOI: 10.1093/nar/gkab225

Response 30: Thank you for your suggestion. We have deleted the website and quoted the references you kindly provided in Line 458.

Point 31: Line 446: Put in reference and remove website

Response 31: Thank you for your suggestion. We have deleted the website and quoted the references in Line 471.

Point 32: Line 456: Put in reference and remove website DOI: 10.1093/nar/27.1.295

Response 32: Thank you for your suggestion. We have deleted the website and quoted the references you kindly provided in Line 481.

Point 33: Line 479: Please cite PRJNA749387 when used in the text

Response 33: Thank you for your suggestion. Not mentioned elsewhere in the text.

Point 34: Line 485-500: This does not match what was introduced. The team needs to spend some time to rewrite the conclusion so that what is suggested in the abstract and introduction match that which they conclude. As it stands it is only half way there

Response 34: Thank you for your suggestion. We have modified the conclusion setion from “In this study, we systematically identified members of the type III PKS gene family in the wheat genome for the first time. The family contains 81 members. Similar to those found in other species, the PKS genes in wheat can be divided into two categories comprising CHS and non-CHS genes. Expansion of the wheat PKS gene family mainly occurred via tandem duplication and segmental duplication events. Synteny analysis showed that three PKS genes in wheat are collinear with rice genes, four with maize genes, and 49 with two grain wheat genes. In addition, we systematically analyzed the physical and chemical properties, chromosome distribution, gene structures, conserved motifs, and promoter elements for 81 TaPKS genes, and constructed a phylogenetic tree. Moreover, we analyzed the expression patterns of PKS genes in different wheat tissues and their responses to infection with Rhizoctonia cerealis. The responses of four PKS genes to treatment with hormones and Rhizoctonia cerealis were analyzed by qRT-PCR. Wheat PKS family genes may regulate the expression of downstream genes through synergistic or inhibitory effects with hormones. In conclusion, wheat PKS genes play important roles in plant resistance to pathogens. Our results provide an important and comprehensive reference framework for further study of the wheat PKS gene family, and a useful basis for screening candidate Rhizoctonia cerealis resistance genes.” to “In this study, we systematically identified members of the type III PKS gene family in the wheat genome for the first time. The family contains 81 members. Similar to those found in other species, the PKS genes in wheat can be divided into two categories comprising CHS and non-CHS genes. Expansion of the wheat PKS gene family mainly occurred via tandem duplication and segmental duplication events. Synteny analysis showed that the PKS genes in wheat and T. dicoccoides are closely related. In addition, by predicting the promoter elements of TaPKS gene, it was found that the promoter region of TaPKS gene contains many hormone regulatory elements. Therefore, we speculate that TaPKS genes may be regulated by hormones. After that, we analyzed the expression pattern of TaPKS genes in the resistant material H83 treated with salicylic acid, methyl-jasmonate, abscission and ethylene by qRT-PCR, which verified the hypothesis that the expression of TaPKS genes was regulated by hormones (salicylic acid, methyl-jasmonate, abscission and ethylene), indicating that TaPKS genes may participate in the process of wheat sheath blight resistance by mediating hormonal changes in response to Rhizoctonia cerealis infection. Previous RNA-seq data showed that the expression of TaPKS genes was tissue-specific and induced by Rhizoctonia cerealis. The results of qRT-PCR confirmed that the expression of TaPKS was induced by Rhizoctonia cerealis. In addition, we believe that that the response times and expression patterns of these TaPKS genes in the different materials were related to the resistance of these materials to wheat sheath blight. In conclusion, wheat PKS genes play important roles in plant resistance to pathogens. Our results provide an important and comprehensive reference framework for further study of the wheat PKS gene family, and a useful basis for screening candidate Rhizoctonia cerealis resistance genes.”

Reviewer 2 Report

The article "Genome-wide analysis of type III polyketide synthases in wheat and possible roles in wheat sheath blight resistance" by X. Geng et al. is a very complete study on the Chalcone-synthase genes in wheat (TaPKS) and its possible role in the resistance against infection with Rhizoctonia cerealis. The design of the stdy is adequate, the materials and methods appropiate and consclusions are significant. Therefore I reccomend this article for publication in IJMS if the following minor points are attended:

1) P. 2, lines 50-54: The reactions catalyzed by chalcone-synthase are not clearly described, the paragraph is confusing and must be re-written.

2) P.3, lines 100-101: What is the function of "non-chalcone function gene families"? Are those cases of "gain of function"? This point must be clarified.

Author Response

Dear editor/reviewer,

Thank you very much for your letter concerning our manuscript entitled “Genome-wide analysis of type III polyketide synthases in wheat and possible roles in wheat sheath blight resistance”. All the comments are valuable and very helpful for revising and improving our paper, as well as the important guiding significance to our researches. We must thank you for taking the trouble to edit the manuscript. We have studied comments carefully and have made correction which we hope meet with approval. Revised portion are marked up using the“Track Changes” function. The main corrections in the paper and the responds to the comments are as flowing. Should you have any questions, please contact us without hesitate.

Point 1: P. 2, lines 50-54: The reactions catalyzed by chalcone-synthase are not clearly described, the paragraph is confusing and must be re-written.

Response 1: Thank you for your suggestion. We have modified this part from "X" to "Y". In order to make it easier for readers to understand the reaction process, we present the catalytic process in a graph, and the specific catalytic process is shown in the Figure 1 below.

Figure 1. Reaction catalyzed by type III PKSs.

Point 2: P. 3, lines 100-101: What is the function of "non-chalcone function gene families"? Are those cases of "gain of function"? This point must be clarified.

Response 2: Thank you for your suggestion. We have added the definitions of CHS and non CHS in the caption of Figure 2, specifically described as” The CHS superfamily includes CHS and a series of chalcone synthase-like (CHSL, also called non-CHS) genes [20-21]. Their functional differences are mainly reflected in the selection of initial substrates and the number of condensation reactions (polyketide chain extension length [8-10].” in Line 110-112.